# Locomotion Outcome Improvement in Mice with Glioblastoma Multiforme after Treatment with Anastrozole

**DOI:** 10.3390/brainsci13030496

**Published:** 2023-03-15

**Authors:** Irene Guadalupe Aguilar-García, Ismael Jiménez-Estrada, Rolando Castañeda-Arellano, Jonatan Alpirez, Gerardo Mendizabal-Ruiz, Judith Marcela Dueñas-Jiménez, Coral Estefania Gutiérrez-Almeida, Laura Paulina Osuna-Carrasco, Viviana Ramírez-Abundis, Sergio Horacio Dueñas-Jiménez

**Affiliations:** 1Departamento de Biología Molecular y Genómica, Centro Universitario de Ciencias de la Salud, Universidad de Guadalajara, Guadalajara 44340, Mexico; 2Departamento de Fisiología, Biofísica y Neurociencias, Centro de Investigación y Estudios Avanzados del Instituto Politécnico Nacional, Ciudad de Mexico 07000, Mexico; 3Laboratorio de Farmacología, Centro de Investigación Multidisciplinario en Salud, Centro Universitario de Tonalá, Universidad de Guadalajara, Tonalá 45425, Mexico; 4Departamento de Neurociencias, Centro Universitario de Ciencias de la Salud, Universidad de Guadalajara, Guadalajara 44340, Mexico; 5Centro Universitario de Ciencias Exactas e Ingenierías, Universidad de Guadalajara, Guadalajara 44430, Mexico; 6Departamento de Fisiología, Centro Universitario de Ciencias de la Salud, Universidad de Guadalajara, Guadalajara 44340, Mexico

**Keywords:** locomotion, glioblastoma, anastrozole

## Abstract

Glioblastoma Multiforme (GBM) is a tumor that infiltrates several brain structures. GBM is associated with abnormal motor activities resulting in impaired mobility, producing a loss of functional motor independence. We used a GBM xenograft implanted in the striatum to analyze the changes in Y (vertical) and X (horizontal) axis displacement of the metatarsus, ankle, and knee. We analyzed the steps dissimilarity factor between control and GBM mice with and without anastrozole. The body weight of the untreated animals decreased compared to treated mice. Anastrozole reduced the malignant cells and decreased GPR30 and ERα receptor expression. In addition, we observed a partial recovery in metatarsus and knee joint displacement (dissimilarity factor). The vertical axis displacement of the GBM+anastrozole group showed a difference in the right metatarsus, right knee, and left ankle compared to the GBM group. In the horizontal axis displacement of the right metatarsus, ankle, and knee, the GBM+anastrozole group exhibited a difference at the last third of the step cycle compared to the GBM group. Thus, anastrozole partially modified joint displacement. The dissimilarity factor and the vertical and horizontal displacements study will be of interest in GBM patients with locomotion alterations. Hindlimb displacement and gait locomotion analysis could be a valuable methodological tool in experimental and clinical studies to help diagnose locomotive deficits related to GBM.

## 1. Introduction

Glioblastoma Multiforme (GBM) is the most aggressive type of glioma [1], with a median survival expectancy of 15–18 months after the diagnosis and a five-year survival rate of <10% [2]. GBM patients’ standard treatment consists of surgical tumor resection, several radiotherapy cycles, and the chemotherapy drug temozolomide. Unfortunately, this combined intervention protocol is ineffective [3]. Therefore, it is essential to find a groundbreaking treatment for GBM [4]. Focal neurological deficits (i.e., motor weakness) typically occur in glioma patients and are associated with growth into motor areas. The striatal area has a significant role in controlling motor activities, and murine striatal glioblastoma models in this area allow the assessment of motor abilities [5]. Several studies involving this region show their participation in neurological disorders associated with abnormal motor activity [6,7]. Likewise, the degeneration in this structure impairs diverse motor and behavioral tasks [8]. However, more longitudinal studies of motor dysfunction in animal models are needed, as well as tools for early detection. The hindlimb displacement in mice walking over-ground has not been studied in murine striatal glioblastoma xenograft and could be an adequate model to test motor alterations.

Glioblastoma is a heterogeneous tumor with multiple redundant intracellular pathways, generating several subtypes [1,9]. Their expression is associated with the patient’s survival outcome [10]. The estrogens directly bind classical or membrane estrogen receptors to initiate gene expression, suggesting diverse functions and tumoral properties. Third-generation aromatase inhibitors, such as anastrozole, have reduced estrogen levels by over 96%. This change is associated with decreased malignant cell viability and tumor growth [11]. This novel strategy should aim to target glioma growth and prevent the functional deterioration of spared brain networks. Based on these premises, we have set up a GBM mouse model by injecting C6 cells into the striatum to monitor locomotive behavioral dysfunction induced by tumor growth. The striatum in murine models is the topographic location showing the densest presence of gliomas. Moreover, the location of the xenograft in the striatum was due to the availability of sensitive behavioral tests that allowed the longitudinal assessment of motor abilities in the same animals. Additionally, this strategy allowed us to count the number of malignant cells and provided us with a new diagnosis tool to correlate tumor growth and hindlimb motor alterations.

## 2. Methods

### 2.1. Cells Culture 

The rat C6 cell line (ATCC, CCL-107^TM^) was cultured in DMEM-F12 high in glucose (Caisson DFL-14), supplemented with 10% fetal bovine serum (Gibco 26140, MO, USA) and 1% penicillin/streptomycin (Corning, 30-002-CL, AZ, USA). The cells underwent incubation at 37 °C in a humidified atmosphere containing 95% air and 5% CO_2_. Afterward, cells were separated from the plate to implant them (1 × 10^6^) in nude/nude mice into the right striatum.

### 2.2. Animals

We housed male Balb-C-nude/nude (Jackson lab: NU/J 002010), 6–7 weeks of age. The animals were kept under sterile conditions in boxes with sterile air exchange and light-dark cycles of 12 × 12 h, with controlled temperature between 23 and 25 °C, and free access to water and food until the day of surgery. All animal experiments were performed following the USA Guide for the Care and Use of Laboratory Animals, National Institutes of Health, The Mexican Regulation of Animal Care and Maintenance (NOM-062-ZOO-1999, 2001), and the institutional University of Guadalajara regulations. 

### 2.3. Glioblastoma Xenograft and Mice Treatment

We formed two groups of mice; both groups received a C6 cells’ xenograft; the first group was not treated (GBM group, *n* = 5), and the other one was treated with anastrozole (GBM+anastrozole, *n* = 5). We anesthetized mice with sevoflurane (3%). We made an incision in the brain midline of the scalp and a small hole in the skull following the stereotaxic coordinates (X = 1.34 mm, Y = 1.5 mm, and Z = 3.5 mm). We administered 1 × 10^6^ cells in 2 μL of DMEM-F12 using a Hamilton syringe in mice’s right striatum (See Appendix A). Anastrozole (Sigma Aldrich A2736, MO, USA) was dissolved in DMSO 0.1 mM to obtain a final concentration of 500 μg/mL (stock solution) and stored at −20 °C. The drug (0.1 mg/kg) was administered through the tail vein with an insulin syringe (0.5 mL daily) for seven days.

### 2.4. Body Weight in Mice

The mice were randomly separated into 2 different groups: the GBM group and the GBM+anastrozole group. Then, they were fed with ad libitum access to food and water. The mice were kept with monitoring of food intake, water intake, and excretion, and were sacrificed at day 14. The body weight initially was 21 g ± 1 g in both groups.

### 2.5. Hematoxylin & Eosin Staining

The animals were anesthetized intraperitoneally with pentobarbital at 160 mg/kg of body weight and sacrificed by intracardiac perfusion using a saline solution (0.9%) and 4% paraformaldehyde. Brains were removed and placed in the same fixed solution at 4 °C. The brains were sectioned in the coronal plane at a thickness of thirty micrometers with a vibratome (Thermo Scientific, HM650V, MA, USA), and then processed for histology by Hematoxylin & Eosin staining. The slices were first submerged for two minutes in water and after three minutes in hematoxylin (Sigma H3136) and then three seconds in acid alcohol (1% HCl in 70% alcohol), washed with distilled water, and immersed in eosin (Sigma Aldrich 212954, MO USA) for a minute and a half before being washed with tap water for thirty seconds. For dehydration, the tissues were put in an increasing gradient of ethanol and xylol: 70% ethanol for 3 s, 90% ethanol for 3 s, and 96% alcohol for 3 min, twice in 100% ethanol for 5 min, and then twice in xylene for 5 min. We used entellan for mounting sections and observed them under a microscope (Carl-Zeiss Aalen, Germany) at 10× and 40×. We counted cells using a 40× objective, considering four fields of the ipsilateral hemisphere. 

### 2.6. Immunofluorescence 

We used immunofluorescence for GFAP, GRP30, and ERα. The brain sections were incubated at room temperature for 30 min in PBS 1x/Triton X-100 0.2%. Next, the tissue sections were incubated for 1 h in PBS 1X bovine albumin serum 1%. Then, the sections were incubated overnight with GFAP antibody (1:750, DAKO, Z0334, RRID: AB_10013382), anti-ERα mouse monoclonal (1:500, Abcam ab 66102 RRID: AB_310305), and anti-GPR30 mouse monoclonal (1:500, Abcam ab 39742 RRID: AB_1950438). Lastly, the secondary antibodies: FITC anti-rabbit IgG (1:500, Jackson AB_2337972) and Alexa fluor 594 polyclonal rabbit (1:1000, Abcam ab150080) were used for a 2 h incubation. We used a 40× oil immersion objective and the Olympus BX51WI microscope.

### 2.7. Tunnel Walk Recordings

We conducted a locomotion analysis studying the metatarsus, ankle, and knee joints’ hindlimbs displacements. We used the dissimilarity factor (DF) and vertical/horizontal displacements of the mice’s strides. We took the data registered before tumor implantation (control group) and after seven days (GBM group), as well as after fourteen days (GBM+anastrozole group) of xenograft implant. We took video recordings while the animals were walking on a transparent Plexiglas tunnel. The video was registered using two synchronized cameras recording left and right hindlimbs simultaneously. We set the cameras to record at 240 fps with a resolution of 1280 × 720 pixels. Post-processing was applied to the resulting videos to remove spherical distortion due to the lenses by estimating a homographic matrix using four points on the image [12]. A step cycle corresponds to when the metatarsus lifts off to when the metatarsus touches down. Using custom-made software, we marked knee, ankle, and metatarsus joints on each video frame for each step. We studied each joint’s displacement curves and values through software developed in our laboratory. Each one of the animal’s steps was captured on the video separately. During several steps, we generated displacement curves on the horizontal and vertical axes concerning time for each joint in the left and right hind limbs. All curves were normalized according to the stride using a value range from one to 100, employing a spline-based interpolation.

### 2.8. Dissimilarity Factor Analysis

We measured the dissimilarity factor (*DF*) to compare the control group steps versus glioblastoma and anastrozole-treated animals to determine the locomotion changes between animal groups. We compared their displacement curves and calculated the dissimilarity factor between them using the Euclidean distance between each of the points of the normalized curve on the horizontal (X) and vertical (Y) axes as
(1)$$DF_{\langle a,b\rangle }=\frac{1}{200}\sqrt{\overset{100}{\underset{i=1}{\sum }}{\left(x_{a}\left(i\right)-x_{b}\left(i\right)\right)}^{2}+\overset{100}{\underset{i=1}{\sum }}{\left(y_{a}\left(i\right)-y_{b}\left(i\right)\right)}^{2}}$$
where *DF*<*a*,*b*> is the squared error between every point of the normalized curves, defined as difference factor (*DF*); “*x_a_ (i)* − *x_b_ (i)*” is the difference (*d*) between the coordinates in *x*, and “*y_a_ (i)* − *y_b_ (i)*” in *y* of every point in the graph, when comparing two steps (*a* and *b*); and “*i*” is the percent in the step cycle.

We compared the curves of every animal in the control and the experimental groups (GBM and GBM+anastrozole). We analyzed the curves of a control animal vs. all control animals and the steep curve of an experimental animal concerning all control animals [Leon-Moreno et al., 2020]. Then, we had these comparations: control vs. control, GBM vs. control, and GBM+anastrozole vs. control. We estimated the *DF* values and analyzed statistical significances with an ANOVA test of unidirectional via and a post hoc Tukey.

### 2.9. Vertical/Horizontal Displacement Analysis

We analyzed the vertical and horizontal displacements separately. We took each joint’s vertical/horizontal displacement data and averaged it per group. The measurement of the hindlimbs displacement of each group was six repetitions, per side, per mouse. Then, we compared the experimental groups (GBM and GBM+anastrozole) versus the control group. We evaluated significant differences at every two perceptual points of the step cycle between groups through a student’s *t*-test (a = 0.05). A locally designed MATLAB script was used for the pattern comparison analysis.

## 3. Statistics

The dissimilarity factors were expressed as means ± SD. We analyzed the data using one-way ANOVA with Tukey post hoc. The data analysis for body weight, cell counting, and horizontal and vertical displacement was performed through an unpaired one-tailed student’s *t*-test, and a *p*-value of * <0.05 was considered statistically significant. We conducted the statistical analysis using the Prism 9.0 software GraphPad and MATLAB R 2021b.

## 4. Results

### 4.1. Body Weight in GBM and GBM+Anastrozole Groups

We evaluated the mice’s weight from the xenograft day until 14 days post-transplantation. During the first 11 days after transplantation, GBM+anastrozole mice maintained a weight between 20 and 22 g (Figure 1). On days 12 and 13, there was no weight loss in the GBM group, while the GBM+anastrozole animals remained unchanged. A significant difference in body weight between the GBM and GBM+anastrozole groups on days 12 and 13 (*p* < 0.05) was observed. 

### 4.2. Histopathological Changes in the Striatal Area of GBM and GBM+Anastrozole Mice

We analyzed the tumor volume of GBM vs. GBM+anastrozole (Figure 2A–C). The anastrozole-treated animals did not show statistical differences in tumor volume reduction at 14 days of treatment of 23.4 mm^3^ ± 2.5, with respect to 27.5 mm^3^ ± 3.2 of tumor volume of GBM (Figure 2F). However, the H&E staining showed that the glioma in mice treated with anastrozole exhibited better-defined tumor margins and fewer invasive cells to the GBM striatum compared with other brain regions. 

Gliomas present typical malignant cell characteristics of humans, such as nuclear atypia and multinucleation. They also exhibited areas of necrosis and palisade arrangement (Figure 2D). The contralateral striatal area showed a normal distribution of glial cells and no angiogenesis (Figure 2E). Compared with the GBM group, the GBM+anastrozole group exhibits fewer cells in the tumor tissue (Figure 2E). Some striatum slices in GBM+anastrozole mice did not show tumor cells. The treatment with anastrozole reduces (19%) the number of glioblastoma cells in the striatum as compared to the GBM group (Figure 2G).

### 4.3. Expression of ERα and GPR30 Receptors in the Study Groups

As shown in Figure 3, the striatal cells in the GBM group present intense ERα-GFAP staining at 14 days post-xenograft (Figure 3A). At the same time, cells in the GBM+anastrozole group exhibited a less intense expression of ERα (Figure 3B). Furthermore, GPR30 immunopositive cells are present in Glioblastoma multiforme. The GBM group shows a highly positive reaction to GPR30 cells, which co-localized mainly in the cell nucleus (Figure 3C). In contrast, anastrozole treatment strongly reduced the GPR30-positive cells in glioblastoma (Figure 3D).

### 4.4. Changes in Mice Locomotion with Glioblastoma and Those Treated with Anastrozole

We analyzed the hindlimb displacement in all study groups and compared dissimilarity factors before and after xenograft in the same animal. We observed a significant effect on the DF of mice 14 days following the xenograft. The left metatarsus DF of the control group had a statistical difference (* *p* = 0.029, Figure 4A) compared to the GBM group. The left metatarsus DF in the GBM vs. GBM+anastrozole mice groups’ curves does not exhibit statistical differences (Figure 4A). In the left ankle, there was no difference between the study groups (Figure 4B). The left knee DF showed statistically significant changes between the GBM and the control group (* *p* = 0.0178). The differences were also present in GBM vs. GBM+anastrozole group (* *p* = 0.0137). There were no differences between the control and anastrozole-treated groups. (Figure 4C). So, there was a recovery in the DF of treated animals.

### 4.5. The Horizontal Displacement among Different Study Groups

The left metatarsus, ankle, and knee horizontal displacement did not show a statistical difference among the study groups. Note that control vs. GBM (*), control vs. GBM+anastrozole (+), and GBM vs. GBM+anastrozole (x) are similar (Figure 5A–C). In contrast, the right metatarsus horizontal displacement shows a statistical difference in GBM vs. GBM+anastrozole group from bins 86 to 100 with a 16% difference (* *p* < 0.05, Figure 5D). The right ankle horizontal displacement showed a statistical difference between GBM vs. GBM+anastrozole (x) groups from the bins 72 to 100 with a 28% difference, and GBM+anastrozole vs. control (+) shows the difference from the bin 68–70 with a 2% difference (Figure 5E). In the right knee, horizontal displacement shows statistical changes in GBM+anastrozole vs. control (+) from bins 70 to 100 with a 32% difference, and in GBM+anastrozole vs. GBM (x), from bins 82 to 100 with a 24% difference (Figure 5F).

### 4.6. Changes in Vertical Displacement 

The left metatarsus did not differ among the studied groups (Figure 6A). In contrast, the right metatarsus exhibited changes in the control group vs. GBM+anastrozole (+) from bins 66 to 72 with an 8% difference, in GBM+anastrozole vs. GBM (x) from bins 56 to 58 with a 4% difference, and also in the GBM vs. control (*) group from bins 50 to 54 with a 6% difference (Figure 6D).

The left ankle vertical displacement showed statistically significant changes between control vs. GBM (*) from bins 60 to 68 with an 8% difference, and GBM compared to GBM+anastrozole (x) from bins 18 to 34 with a 16% difference (*p* < 0.05 * Figure 6B). The right ankle showed a difference between control and GBM+anastrozole from bins 58 to 70 with a 12% difference (Figure 6E). 

The left knee vertical displacement between control vs. GBM+anastrozole groups changed from bins 50 to 52 with a 4% difference (Figure 6C). The right knee vertical displacement between GBM+anastrozole vs. GBM groups changed from bins 52 to 62 with a 12% difference, GBM+anastrozole vs. control from bins 20 to 34 with a 14% difference, and GBM vs. control from bins 26 to 28 and 46 to 52 with an 8% difference (* *p* < 0.05, Figure 6F).

## 5. Discussion

The body weight loss in animals treated with anastrozole occurred in rats [13,14] and in the transgenic female 3xTgAD mice [15]. This work described changes in body weight, with mice maintaining their weight on days 12 and 13. Breast cancer patients commonly report weight gain after tamoxifen or aromatase inhibitor administration [16]. Thus, we suggest that anastrozole could contribute to maintaining weight through steroid regulation of body weight mass. 

The histopathological characteristics of the GBM developed in the striatum are similar to those observed in patients with GBM. There is tissue necrosis, neovascularization, and an arrangement palisade pattern of the tumor cells [17]. Glioma cells are reduced on the 14th post-treatment day, indicating an anastrozole antiproliferative and apoptotic effect. Such results agree with work reported for lung and breast cancers [18,19].

We observed a qualitative decrease in the expression of estrogen receptors (ERα and GPR30) in tumor GBM+anastrozole xenografted tissue. An increase in estrogen and its receptors is associated with tumor growth in different cancer types [20]. The increase in ERα expression was related to reduced GBM patient survival [21]. Clinical trials have shown that anastrozole is better than selective estrogen modulators against breast cancer [22,23]. This effect is due to a systemic reduction of 17ß-estradiol and negative ERα expression regulation [24]. The recently discovered estrogen receptor GPR30 is present in several cancer cells [25]. The expression of this receptor plays an essential role in the tumor growth of gastric cancer [26], breast cancer [27], and endometrial cancer [28], among others. This study found that ERα and GPR30 decreased expression in the GBM anastrozole-treated group. The reduced number of tumor cells could be due to low estrogen alpha and GPER receptor expression and estrogen levels [10,29]. Further experiments are needed to quantify estrogen receptor expression in GBM-treated anastrozole mice.

Brain tumors are related to cognitive and motor deficits [30,31]. Brain tumors significantly affect motor networks due to alterations in cortical areas [32,33]. They also affect functional connectivity between cortical and subcortical motor areas [34]. A study reported important aspects concerning tumor growth evaluation and specific motor behavioral alterations, particularly gait instability, in a rat model [35]. In clinical studies of GBM, gait instability is a common motor symptom caused by tumor invasion [36,37]. GBM models in rats show regions of focal invasion into brain tissue, similar to the diffuse infiltrating pattern seen in GBM patients [38]. Glioblastoma growth into motor areas is associated with an alteration in gait locomotion. A critical area related to locomotion is the striatum, and modifications in this area may produce complications in the rhythmic alternation of limbs [39]. The striatum modulates treadmill locomotion in rats and humans during free walking [6,40]. The GBM can produce diverse motor alterations because it has no defined limits. 

In our experiments, we observed a reduction in the dissimilarity factors and, consequently, an improvement in the left hindlimb metatarsus and knee displacement in GBM+anastrozole mice. Thus, anastrozole improved gait locomotion, probably due to a brain tumor reduction in the right motor area (Figure 4). Concerning the step cycle changes, the horizontal and vertical displacement of the right side showed differences between the study groups. GBM tumor growth in the striatum may lead to impaired hindlimb displacement and motor impairment in each step cycle [37]. Further studies are needed to establish the relationship between lesion location resulting in the effects of tumors, or the pathways producing changes in the spinal cord central pattern generators [41]. Following the motor deficit found in our experiments, treatment and rehabilitation will depend on previous treatments, tumor area swelling, and invasivity.

The locomotor system comprises centers in the brainstem controlling spinal circuitry. The motor deficits of several hindlimb joint displacements could be attributed to the dispersed commands to engage the joint generator circuits [42,43].

In mice walking over-ground, we found changes in the right metatarsus, ankle, and knee joints after anastrozole. It seems possible that GBM progression and regression occur differentially in the right and left brain stems. Further studies are necessary to evaluate the recovery of brain stem–spinal cord pathways and their relationship with tumor size reduction. It will be interesting to study why anastrozole reduces the horizontal displacement in the right hindlimb. Reducing the step cycle’s variability could help stabilize locomotion and navigation. The hindlimbs’ right gait compensation could stabilize the correct hindlimb gait.

Additionally, our study showed that the alterations in vertical displacement were more dispersed than horizontal displacement in both hindlimbs. The neural pathways that activate the displacement are not known, and neither are the tumor dimensions nor the precise infiltration. Anastrozole produces differential effects in the tumoral cells. We need to study the spatial tumor dimensions to propose an anastrozole effect to obtain a clear conclusion. 

An important finding is that the vertical axis of the displacement of GBM+anastrozole is similar to the control group, which implies that anastrozole regulates the changes due to GBM in the ankle joint displacement.

The pyramidal pathways project to motor neurons and the CPG (Central Pattern Generator). They adapt the basic locomotor pattern to environmental constraints [44,45]. They could participate in adapting the motor system to the brain stem alterations produced by the tumor. 

In this study, the variations found in horizontal and vertical displacements in the different joints suggest independent burst pattern generators for each joint of both hindlimbs on several projections coming from the brain stem. 

## 6. Conclusions

We addressed the functional relevance of the antineoplastic effect of anastrozole treatment by regulating the ERα and GPR30 expression in GBM xenograft. Thus, anastrozole partially recovered joint displacement by modifications in vertical and horizontal displacements in different phases of the step cycle. It will be interesting to study whether similar results occur in some patients with GBM exhibiting locomotion alterations. Hindlimb displacement and gait locomotion analysis could be a valuable methodological tool in experimental and clinical studies to develop new therapeutic approaches against locomotive deficits produced by GBM.

## Figures and Tables

**Figure 1 brainsci-13-00496-f001:**
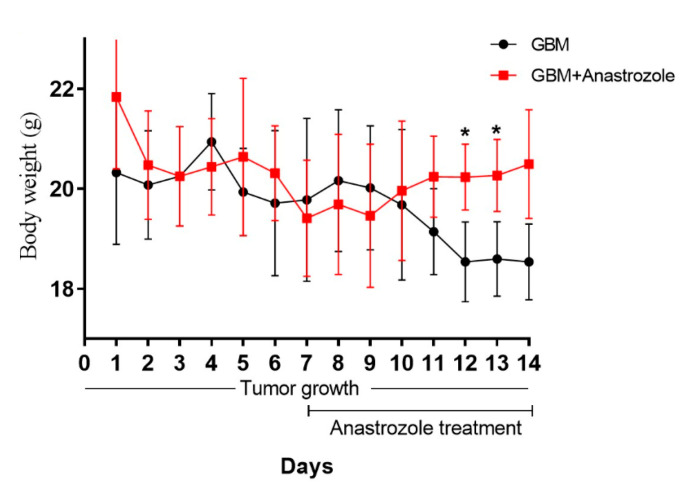
Body Weight in mice. Graph illustrating the body weight changes of mice monitored 14 days after xenograft. Data correspond to GBM and GBM+anastrozole. The GBM+anastrozole group showed a significant increase in body weight on the 12th and 13th days compared to the GBM group. The data show Mean ± SE values. The asterisks indicated statistical differences between groups (Mann–Whitney U test; *p* < 0.05).

**Figure 2 brainsci-13-00496-f002:**
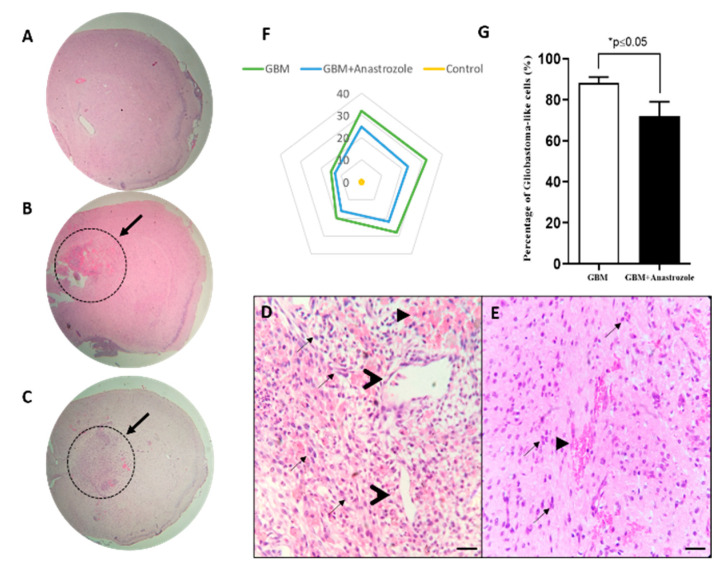
Histopathological changes in Striatum. (**A**) Photograph illustrating a transverse area in the right striatum of an untreated GBM mouse. (**B**,**C**) Administration of anastrozole does not reduce tumor growth in the mice glioma model. Glioblastoma tumor tissue shows morphological features that include a disordered arrangement of clear and large cells with condensed nuclei and darkly stained cytoplasm. Arrows point to necrotic centers indicating areas of necrosis. (**D**,**E**) Photograph of an area in the striatum of a GBM+anastrozole mouse showing the arrangement of cells. Arrows head indicates vessels.Arrows without line shows necrotic area. Note that it contains fewer large cells and a reduced number of nuclei than the striatal GBM tissue. (**F**) The tumor volume was measured in mm^3^. (**G**) Graph exhibiting data of the counted tumor cells in GBM and GBM+anastrozole groups. Bars represent mean ± SD (*n* = 5 animals). The asterisk indicates statistical differences between groups (*t*-student test; *p* < 0.05). Scale bar = 50 µm and 200 µm.

**Figure 3 brainsci-13-00496-f003:**
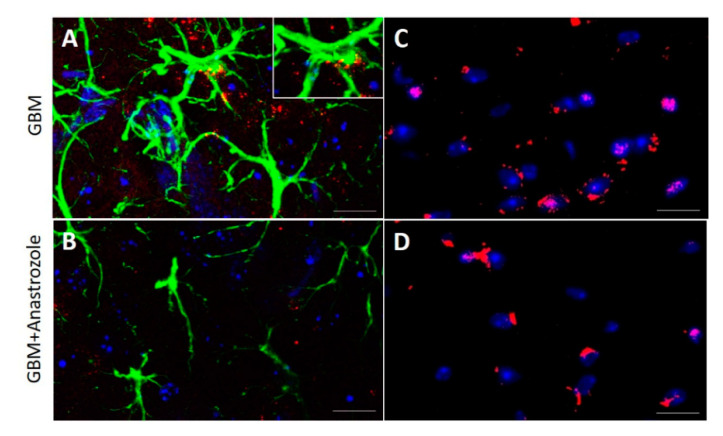
Expression of ERα and GPR30 immunopositive C6 cells. (**A**) Microphotograph showing the merge of GFAP immunopositive cells (green), ERα expression (red), and cell nuclei stained with DAPI (blue) in striatal tissue of GBM. (**B**) In GBM+anastrozole mice, striatal cells and striatal tissue exhibited a decrease in staining to ERα in GFAP and DAPI. The insert clearly shows GFAP-immunofluorescence with ERα co-expression in glioblastoma cells. (**C**,**D**) The microphotographs show GPR30 expression (red) and nuclei (blue) in striatal tissue slides of GBM and GBM+anastrozole animals, respectively. GBM tissue exhibits more nuclei and higher GPR30 expression than those observed in GBM+anastrozole tissue. Scale bar = 30 µm.

**Figure 4 brainsci-13-00496-f004:**
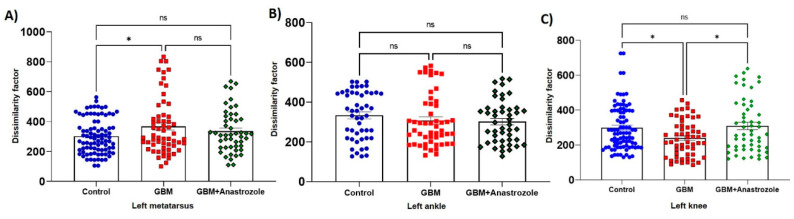
Dissimilarity factor changes in metatarsus, ankle, and knee of the left hindlimb in GBM-control and GBM+anastrozole groups. (**A**) The dissimilarity factor (DF) in the left hindlimb metatarsus has a statistical difference between the control and GBM groups (* *p* < 0.027). (**B**) The DF in the ankle did not show differences among the control, GBM, and GBM+anastrozole groups. (**C**) The DF exhibited a significant difference between control versus GBM (* *p* < 0.0178) and GBM versus GBM+anastrozole (* *p* < 0.0137). The data show Mean ± SD values. The asterisks indicated statistical differences between groups using an ANOVA test.

**Figure 5 brainsci-13-00496-f005:**
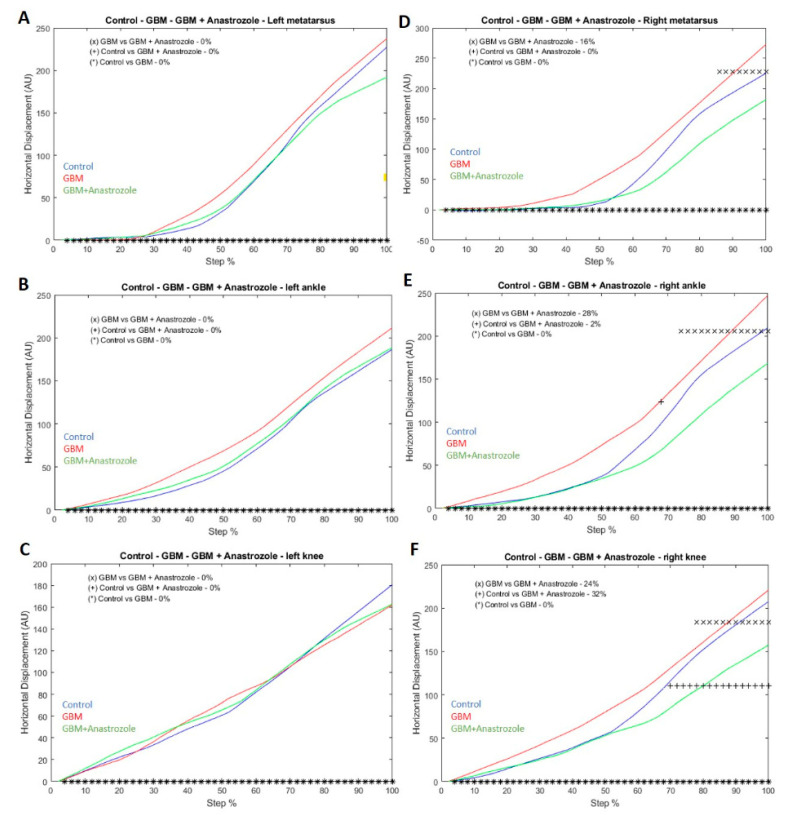
Left and right hindlimb metatarsus, ankle, and knee displacement in the horizontal axis. The left metatarsus, ankle, and knee horizontal displacement does not show a statistical difference among the control and GBM groups. (**A**–**C**). The right metatarsus horizontal displacement significantly changed in GBM vs. GBM+anastrozole groups (* *p* < 0.05, (**D**)). The right ankle horizontal displacement showed significant changes in the step cycle in the GBM vs. GBM+anastrozole groups (**E**). The right knee horizontal displacement showed significant changes among the GBM and GBM+anastrozole groups, and the GBM+anastrozole versus control groups (* *p* < 0.05, (**F**)). The symbols (*, +, x) over zero (0) indicate statistical differences between groups (student test; *p* < 0.05). Every symbol corresponds to two bins.

**Figure 6 brainsci-13-00496-f006:**
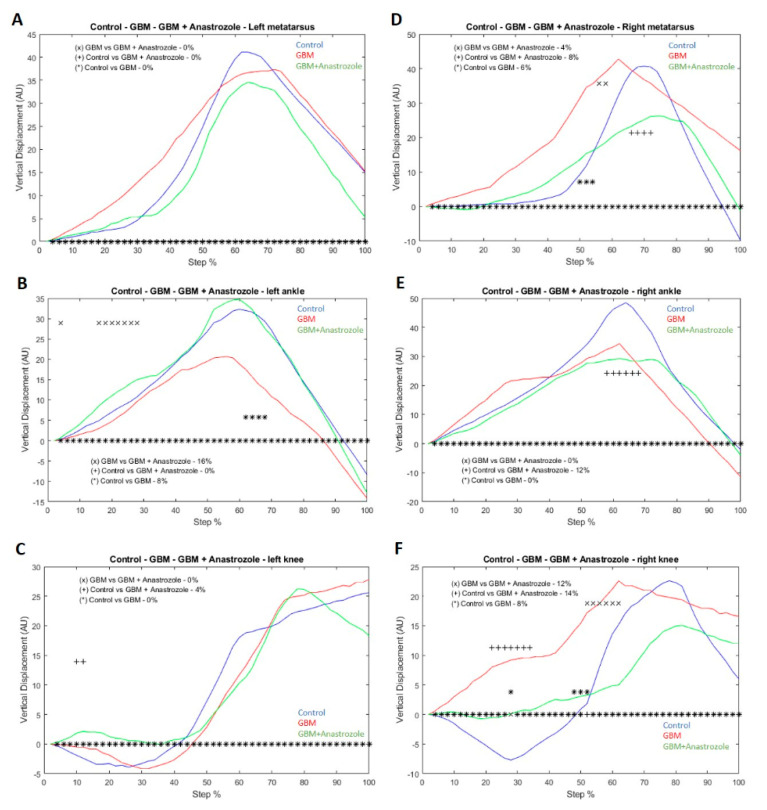
Left and right hindlimb metatarsus, ankle, and knee displacement in the vertical axis. The left metatarsus vertical displacement was not statistically significant among groups during the step cycle (**A**). The left ankle vertical displacement shows significant changes in some periods of the step cycle (**B**). It occurred between control versus GBM group and GBM+anastrozole versus GBM (* *p* < 0.05). The left knee vertical displacement changes significantly at the beginning of the step cycle. It appeared between control and GBM+anastrozole groups (**C**). The right metatarsus shows a statistically significant difference in various bins of the step cycle. The changes were observed in GBM versus GBM+anastrozole group, GBM+anastrozole versus control, and control versus GBM (* *p* < 0.05) (**D**). The right ankle vertical displacement showed a statistically significant change in the middle of the step cycle. It occurred between the control versus GBM+anastrozole groups (* *p* < 0.05) (**E**). The right knee vertical displacement shows a statistical difference in several parts of the step cycle. It occurred between the GBM and the GBM+anastrozole groups, control versus GBM+anastrozole, and control versus GBM groups (* *p* < 0.05) (**F**). The symbols (*, +, x) over zero (0) indicated statistical differences between groups (student test; *p* < 0.05). Every symbol above zero corresponds to two bins with a significant statistical difference.

## Data Availability

Not applicable.

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
