# Peer review of "Locomotion Outcome Improvement in Mice with Glioblastoma Multiforme after Treatment with Anastrozole"

_brainsci, 2023, doi:10.3390/brainsci13030496_

Round 1

Reviewer 1 Report

Comments and Suggestions for Authors

The manuscript brainsci-2270366 entitled “Locomotion outcome improvement in mice with Glioblastoma Multiforme after treatment with anastrozole», investigated the Improvement in locomotion in mice with glioblastoma multiforme following anastrozole treatment. They have used a GBM xenograft implanted in the striatum to analyse the changes in Y (vertical) and X (horizontal) axis displacement of the metatarsus, ankle, and knee.

The authors showed that Anastrozole reduced the malignant cells and they observed a partial recovery in metatarsus and knee joint displacement

In general, the study is interesting overall, and the findings may offer some additional information.

Furthermore, I have some notable concerns that should be taken in consideration

for the material and methods section, the authors should provide details on the initial body weight of mice used per group.

The authors should have also indicated the number of repetitions that were made for each animal for the analysis studying the metatarsus, ankle, and knee joints hindlimbs displacements

In terms of form, the authors should respect the classical order of the redaction of a scientific article: this is why I insist on adding a conclusion which reflect well the findings.

Except for these three remarks, I find that this study has a very interesting scientific value.

Author Response

We so grateful for your patience and dedication in reviewing the manuscript. According to the observations sent to you, the changes have already been made to the paper.

Response to Reviewer 1 Comments

Reviewer #1: The manuscript brainsci-2270366 entitled “Locomotion outcome improvement in mice with Glioblastoma Multiforme after treatment with anastrozole», investigated the Improvement in locomotion in mice with glioblastoma multiform following anastrozole treatment. They have used a GBM xenograft implanted in the striatum to analyze the changes in Y (vertical) and X (horizontal) axis displacement of the metatarsus, ankle, and knee.

The authors showed that Anastrozole reduced the malignant cells and they observed a partial recovery in metatarsus and knee joint displacement

In general, the study is interesting overall, and the findings may offer some additional information.

Furthermore, I have some notable concerns that should be taken in consideration

1. For the material and methods section, the authors should provide details on the initial body weight of mice used per group.

Response 1: We modify that statement in Methods

  1. The authors should have also indicated the number of repetitions that were made for each animal for the analysis studying the metatarsus, ankle, and knee joints hindlimbs displacements

Response 2: We modified those statements indicating the number of repetitions that were made for each animal

3. In terms of form, the authors should respect the classical order of the redaction of a scientific article: this is why I insist on adding a conclusion which reflect well the findings.

Response 3: We extended that statement and added a solid conclusion

Reviewer 2 Report

Comments and Suggestions for Authors

the Authors present the results of a well-conducted study which investigates locomotion outcome improvement in a murine model of GBM after treatment with anastrozole.

As a personal remark, I am somehow against animal studies (especially when unnecessary), but in this case I see no special concerns, or at least no more concerns than I usually have, and I did my best to review the paper in an unbiased manner.

And my impression of the study and the paper is indeed quite positive. I have only one major remark. The use of anastrozole in this setting can be an interesting example of drug repurposing (see for instance Abbruzzese C, Matteoni S, Signore M, Cardone L, Nath K, Glickson JD, Paggi MG. Drug repurposing for the treatment of glioblastoma multiforme. J Exp Clin Cancer Res. 2017 Nov 28;36(1):169. doi: 10.1186/s13046-017-0642-x. PMID: 29179732; PMCID: PMC5704391; again as a personal disclaimer, I am not among the Authors of the paper, but I personally know the group who authored it). Can the Authors expand on the potential clinical relevance of their findings, of course after proper studies in humans?

One minor comment: p<0.0178, is that correct (and similar p-values). Or is it p=0.0178?

Author Response

We so grateful for your patience and dedication in reviewing the manuscript. According to the observations sent to you, the changes have already been made to the paper.

Reviewer #2: The Authors present the results of a well-conducted study which investigates locomotion outcome improvement in a murine model of GBM after treatment with anastrozole.

As a personal remark, I am somehow against animal studies (especially when unnecessary), but in this case I see no special concerns, or at least no more concerns than I usually have, and I did my best to review the paper in an unbiased manner.

  1. And my impression of the study and the paper is indeed quite positive. I have only one major remark. The use of anastrozole in this setting can be an interesting example of drug repurposing (see for instance Abbruzzese C, Matteoni S, Signore M, Cardone L, Nath K, Glickson JD, Paggi MG. Drug repurposing for the treatment of glioblastoma multiforme. J Exp Clin Cancer Res. 2017 Nov 28;36(1):169. doi: 10.1186/s13046-017-0642-x. PMID: 29179732; PMCID: PMC5704391; again as a personal disclaimer, I am not among the Authors of the paper, but I personally know the group who authored it). Can the Authors expand on the potential clinical relevance of their findings, of course after proper studies in humans?

Response 1: Anastrozole is an FDA-approved drug against breast cancer. Studies in patients found aromatase expression in astrocytoma to be higher in glioblastoma. Therefore, the use of an aromatase inhibitor could be helpful in the treatment of glioblastoma.

  1. One minor comment: p<0.0178, is that correct (and similar p-values). Or is it p=0.0178?

Response 2: We modified these statement in the text, it is p=0.0178